# Analysis of Nuclear DNA Content and Karyotype of *Phaseolus vulgaris* L.

**DOI:** 10.3390/genes14010047

**Published:** 2022-12-23

**Authors:** Haluk Kulaz, Solmaz Najafi, Ruveyde Tuncturk, Murat Tuncturk, Marzough Aziz Albalawi, Adel I. Alalawy, Atif Abdulwahab A. Oyouni, Abdulrahman Alasmari, Peter Poczai, R. Z. Sayyed

**Affiliations:** 1Department of Field Crops, Faculty of Agriculture, Van Yuzuncu Yil University, 65090 Van, Turkey; 2Department of Chemistry, University College at Alwajh, University of Tabuk, Tabuk 71491, Saudi Arabia; 3Department of Biochemistry, Faculty of Science, University of Tabuk, Tabuk 71491, Saudi Arabia; 4Genome and Biotechnology Unit, Faculty of Sciences, University of Tabuk, Tabuk 71491, Saudi Arabia; 5Department of Biology, Faculty of Sciences, University of Tabuk, Tabuk 71491, Saudi Arabia; 6Finnish Museum of Natural History, University of Helsinki, FI-00014 Helsinki, Finland; 7Department of Microbiology, PSGVP Mandal’s S I Patil Arts, G B Patel Science and STKV Sangh Commerce College, Shahada 425409, India

**Keywords:** cytological methods, flow cytometry, karyotypic formula, nuclear DNA, *Phaseolus vulgaris* L.

## Abstract

The common bean (*Phaseolus vulgaris* L.), whose annual production is 26 million tons worldwide, is one of the main sources of protein and is known as one of the most important food sources. In this study, the karyotype variations and the genome size of four common bean genotypes in Turkey were investigated to determine whether the geographic variables in these regions affected the genome size and the karyotype parameters. In addition, it is known that as that the cytological and chromosomal parameters change under the influence of the climatic conditions of each region, appropriate and stable cytological methods for each plant facilitate and enable the determination of the chromosomal structure and the identification of specific chromosomes in the genotypes of the relevant region. Correct and valuable information such as this enables breeders and researchers to determine the correct shape and actual size of chromosomes. The genome size of the genotypes was measured with a flow cytometer, and chromosome analyses were performed with the squash method. For each genotype, the karyotype parameters, such as the number of somatic chromosomes, the Mean Total Chromosome Length (MTCL), the Mean Centromere Index (MCI), and the Mean Arm Ratio (MAR), were measured. The results showed that the highest and the lowest amounts of DNA per nucleus (3.28 pg and 1.49 pg) were observed in the Bitlis and Elaziğ genotypes. In addition, all genotype chromosome numbers were counted to be 2n = 2x = 22. The Mean Total Chromosome Length varied from 15.65 µm in Elaziğ to 34.24 µm in the Bitlis genotype. The Mean Chromosome Length ranged between 1.42 µm and 3.11 µm in the Elaziğ and Bitlis genotypes. The Hakkari and Van genotypes consist of eleven metacentric chromosomes, while the Bitlis and Elaziğ genotypes consist of ten metacentric chromosomes and one sub-metacentric chromosome. However, the Mean Centromere Index and Arm Ratio differed considerably among the genotypes. The highest (46.88) and the lowest (43.18) values of the Mean Centromere Index were observed in the Hakkari and Elaziğ genotypes, respectively. On the other hand, the lowest (1.15) and the highest (1.36) values of the Mean Arm Ratio were obtained in the Bitlis and Elaziğ genotypes, respectively. Eventually, intraspecies variations in genome size and chromosomal parameters were observed, and it was determined that the changes in nuclear DNA content and different chromosomal parameters among the four *Phaseolus* genotypes from four different regions of Turkey indicate the effect of climate change in the regions on these parameters. Such information in these areas can be used as useful information for the improvement of this plant and breeding programs.

## 1. Introduction

Legumes are among the most important sources of protein in the diet of many people in developing countries and are the second largest source of human food after cereals [1]. The common bean (*Phaseolus vulgaris*) is one of the most important legumes worldwide, and it is an important source of nutrients, especially in East Africa and Latin America [2]. It ranks first among legumes in terms of its economic value and cultivated areas [3]. According to the Food and Agriculture Organization (FAO), the area for the cultivation of this plant in the world in 2014 was about 26.5 million hectares, with an average yield of 697 kilograms per hectare [4]. Due to the deterioration of the global climate and the reduction in annual rainfall, various stresses, such as drought, salinity, flooding, pests, and diseases, reduce the global yield of crops [5]. The region encompassing Ecuador and Northern Peru is considered the origin of the common bean [6], which has subsequently been dispersed both northwards and southwards due to the establishment of the Mesoamerican and Andean gene pools, respectively [7]. The divergence of the gene pools occurred before the domestication events within the individual gene pools [8,9]. After the independent domestication events, local adaptation created diverse landraces [10], which may have possibly caused morphological and genetic variability. Since the common bean is an important nutrient, it has economic importance; therefore, breeding efforts have focused on the global development of bean species with higher yields by increasing resistance to biotic and abiotic stresses [11]. 

One-third of the world’s arable land suffers from insufficient water for agriculture, and this problem is expected to become more severe with climate change and population growth. Therefore, one of the most essential solutions to combat stress is identifying plants tolerating these conditions with optimal performance and studying their tolerance mechanisms [12]. The lack of tolerance to quantitative abiotic stresses and direct measurement methods makes identifying resistant genotypes challenging. Still, grain yield under normal conditions and stresses seems to be the first step in selecting genotypes for use in breeding work under stress conditions [13]. Determining the correlation between chromosomal characteristics and plant phenotypic traits, and determining the cause-and-effect relationships between them allow the facilitator to choose the most appropriate and logical ratio between components that leads to higher performance [14]. 

One of the most important stages of eugenics studies is the study of genetic diversity and knowledge of chromosomal and genomic characteristics, which are necessary to help these studies. Many classical and molecular cytogenetic studies have been performed on this plant, but since these characteristics are variable under the influence of geographical and ecological factors [15,16], it seems necessary to study the characteristics of the nuclear and chromosomal genomes separately in each region. Several cytogenetic studies have been conducted to investigate the chromosomal structure of the common bean, which include chromosome analysis and cytogenetic maps [17,18,19,20,21,22,23,24,25,26,27,28,29,30]. Genome size information is an important issue in ploidy analysis, genome analysis, taxonomy, evolution, and breeding studies [31,32,33,34,35].

The Eastern Anatolia region is Turkey’s coldest and highest region, and especially under high-altitude conditions, such as those in Van, Muş, Bingöl, Bitlis, and Hakkari, the vegetation period, which is considered to be between the last spring frosts and the first autumn frosts, is quite short. For this reason, it is difficult to establish a reliable culture with long-maturity bean varieties, and production is at risk, so for more reliable bean production under similar ecological conditions, improved cultivars resistant to low temperatures with short maturity periods are needed in this region. A total of 40 common bean genotypes, which were selected among 414 genotypes in previous research conducted by one of the current study’s authors [36] in Turkey, were used in this study.

After the 40 genotypes selected according to international standards were put at our disposal in this geographical climate, it was planned to select the top 4 genotypes (the top genotype from each province, for a total of 4 genotypes) to perform some cytogenetic analyses. 

The purpose of this study was to determine the cytogenetic diversity among the four common bean genotypes, including Elaziğ (E.L.), Bitlis (B.T.), Van (V.N.), and Hakkari (H.K.), that are more suitable for the climate of this region in terms of chromosomal parameters and nuclear DNA content and that could be used in the next breeding programs in this region.

## 2. Materials and Methods

### 2.1. Plant Material Preparation

The common bean (*Phaseolus vulgaris*) genotypes selected in the previous research study were used in this research. In the previous research study [36], the authors collected different genotypes from the selected villages of 8 cities, including Bingöl, Bitlis, Elaziğ, Hakkari, Malatya, Muş, Tunceli, and Van, which are located in the south of the Eastern Anatolia region and engage in bean farming. A total of 414 bean genotypes were harvested in autumn 2009 and spring 2010 and were planted in 2010 and 2011 in Van-Gevaş under farming conditions, and the characteristics of the genotypes were evaluated. International Plant Genetic Resources Institute (IPGRI) and European Union Community Plant Variety Office (EU-CPVO) criteria were used in the characterization of dwarf and climbing bean genotypes. The top four genotypes were selected based on good quantitative, qualitative, biological, and agronomic characteristics, including seed shape, seed color, yield, seed quality, flowering status, and appearance. A genotype name was given to each accession based on the first two letters of the provinces and then the number of the village in which they were collected. Three replications of each genotype were used in the cytogenetic analyses. The study was carried out in Field Crops Department, Agriculture Faculty of Van Yuzuncu Yil University, Van, Turkey, during 2015–2016. All of the common bean accessions analyzed in this study are listed in Table 1.

### 2.2. Karyotype Analysis

The root tip meristems (1–2 cm length) were cut from the seedlings and pretreated with 8-Hydroxiquinolin solution; then, they were fixed in Lewitsky solution (1% chromic acid and 10% formaldehyde) for 36 h at 4 °C, followed by rinsing in distilled water, hydrolyzing in 1N HCl at 60 °C (hydrolysis was performed in a hot water bath) for 10 min, and staining with %2 Aceto-Orcein. Ten metaphase plates were analyzed with the squashing method. Several chromosomal features, including the number of somatic chromosomes, the MTCL (Mean Total Chromosome Length), the MCL (Mean Chromosome Length), the MCI (Mean Centromere Index; CI = Total Length of Short Arms/Total Chromosome Length × 100), the MAR (Mean Arm Ratio; AR = the ratio of the length of the largest to the smallest chromosome), the karyotype formula, and the total diploid chromosome length, were determined in each genotype using microscopic photos. The type of chromosome was determined according to Levan’s method [37].

### 2.3. Determination of the Genome Size

The genome size of the accessions was determined using the FCM. For this purpose, the root tips were fixed as previously described, rinsed with SO and distilled water, and squashed in acetic acid (45%), followed by freezing and dehydration in ethanol (96% and 100%, respectively) for 5 min. A total of 75–100 telophase nuclei were selected for measuring the DNA content in each genotype using the flow cytometry technique with a PARTEC Flow cytometer. The nuclear DNA amount was calibrated using Allium cepa root tips as the control [38]. 

### 2.4. Statistical Analysis

To test the statistical significance of the differences between the genome sizes of the accessions, variance analyses and Duncan tests were performed.

## 3. Results

### 3.1. Chromosome Counting and Analysis

As a result of the chromosome analysis, it was determined that all of the accessions had 22 chromosomes (2n = 2x= 22). The MTCL values varied from 15.65 ± 0.99 µm in E.L. (Elaziğ) to 34.24 ± 0.35 µm in the B.T. (Bitlis) genotype, respectively. In addition, the MCL values ranged between 1.42 ± 0.10 µm in E.L. (Elaziğ) and 3.11 ± 0.32 µm in the B.T. (Bitlis) genotype (Table 2). The H.K. and V.N. genotypes consisted of eleven metacentric chromosomes, while the B.T. and E.L. genotypes consisted of ten metacentric chromosomes and one submetacentric chromosome. The genotypes showed considerable variations in the Centromere Index (CI) and Arm Ratio (AR) values. The highest and lowest CI values (46.88 ± 0.91 and 43.18 ± 0.99) were obtained in the H.K. and E.L. genotypes, respectively. On the other hand, the lowest and the highest AR values (1.15 ± 0.10 and 1.36 ± 0.07) were observed in the B.T. and E.L. genotypes, respectively (Table 2). In addition, the total diploid chromosome length was measured to be from 31.30 ± 0.019 µm in E.L. to 68.48 ± 0.012 µm in B.T. (Table 3).

### 3.2. Estimation of DNA Contents in Genotypes

The genome size of the accessions was determined using the FCM. The mean genome size of the common bean accessions varied between 1.49 ± 0.032 pg in the E.L. genotype and 3.28 ± 0.101 pg in the B.T. genotype (Table 3). 

The variance analysis of the MTCL (Mean Total Chromosome Length) in the four studied genotypes of the common bean (Table 4) showed a statistically significant difference among the studied genotypes at the level of 0.1. The statistical grouping of the average TCL (Total Chromosome Length) in the examined genotypes (Table 5) showed that the genotypes could be divided into three separate groups: genotypes B.T. and H.K., with the highest value of Mean Total Chromosome Length in one group, V.N. in the second group, and the E.L. genotype placed separately in another group.

The results of the variance analysis of the amount of nuclear DNA in the four studied genotypes of the common bean (Table 6) showed that there was a statistically significant difference among the studied genotypes at the level of 0.1. The statistical grouping of the average amount of nuclear DNA in the examined genotypes (Table 7) showed that the genotypes were divided into two separate groups: genotypes B.T., H.K., and V.N., with the highest amount of nuclear DNA, in one group and the E.L. genotype placed separately in another group.

## 4. Discussion

The chromosome number in all genotypes was 2n = 2x = 22, confirming several previous results [17,19,20,21,22,23,24,27,28,29,30]. The MTCL (Mean Total Chromosome Length) recorded in the studied genotypes varied from 15.65 ± 0.99 to 34.24 ± 0.35 micrometers (Table 2), which is almost similar to the previous study on nine genotypes of the common bean, who’s MTCL was measured to be between 13.74 and 29.50 micrometers [24]. In another study on bean karyotype analysis, the MCI (Mean Centromere Index) ranged between 42 and 48, which is very similar to our results (43.18 ± 0.99 to 46.83 ± 3.52) (Table 2) [39]. The MAR (Mean Arm Ratio) was calculated to be between 1.14 ± 0.10 and 1.36 ± 0.07 (Table 2) in this study, which does not correspond to previous results (3.50 ± 0.36 to 4.30 ± 0.35) [24]. The mean genome size of the common bean accessions varied between 1.49 ± 0.032 pg in the E.L. genotype and 3.28 ± 0.101 pg in the B.T. genotype (Table 3). Previous studies showed the genome sizes of the common bean to be 3.7 pg [17], 2.7 pg [18], 1.32 pg [20], 1.40–1.53 pg [40], 1.39–1.41 pg [21], 1.58 pg [41], and 2.65 and 4.96 pg [24]. The results obtained from the current study area similar to some of these results and different from others. This may be due to the use of different methods, internal standards, and accessions or technical problems [42]; in many cases, these changes are related to different geographical and climatic conditions [29].

The variance analysis results showed a significant difference among the investigated genotypes in terms of the measured traits. The greater the diversity of the traits is, the more the selection based on them leads to a better selection response [43]. According to Table 5, it can be observed that there was a statistically significant difference among the studied genotypes (*p* < 0.01) in TCL (Total Chromosome Length). The comparison of the average TCL (Total Chromosome Length) values in the studied genotypes (Table 5) showed that the genotypes could be divided into three separate groups: genotypes B.T. and H.K., with the highest values of Mean Total Chromosome Length, could be put in one group; the V.N. genotype was in the second group; and the E.L. genotype was placed separately in the third group. This indicates the high diversity among the studied genotypes in terms of this parameter. Using this parameter in breeding programs could help to achieve suitable genotypes that can adapt to different environmental conditions [44]. Phenotypic changes always represent a part of the overall diversity, but genotypic changes represent a part of the diversity and heritable changes that increase the response to selection [45].

There was a significant difference in the relative amount of nuclear DNA (*p* < 0.01), which indicates high intra-species diversity among the studied genotypes of the common bean collected from different regions of Turkey. The average comparison (Table 7) showed that the genotypes could be placed in two separate groups.

The amount of DNA in the cell nucleus positively correlates with various cellular parameters, including total length and chromosome volume in the metaphase of mitosis and meiosis. In addition, the C-value of DNA in plants is positively related to the characteristics that interact for determining the growth rate and the type of plant life cycle [46]. Genome size refers to the DNA content in the nucleus and is known to be associated with the nucleus, cell size, division rate, and thus various organism-level traits, such as metabolism, body size, or developmental rate [47]. The C-value is the amount of DNA in the haploid un-replicated nucleus (haploid genome size) [48]. It is typically measured in picograms (pg) for mass or as the total number of nucleotides in megabase pairs (Mbp), where 1 pg is equal to 978 Mbp of DNA [42]. The genome size is mainly estimated using two cytogenetic methods: flow cytometry and Feulgen micro-densitometry [49,50,51,52]. Previous studies have shown that the high probability of genetic variation among species occurs due to intra-species changes in the nuclear DNA amount [45]. In addition, the correlation between genome size, and growth, development characteristics, and climate characteristics has been observed [53]. The difference in genome size is one of the most important evolutionary processes in plants. Many studies have demonstrated that a significant difference in genome size is correlated with the evolution of species and their grouping based on ecological conditions and geographical origin [54]. In breeding programs, the cytogenetic study is considered the first step, since crossing between species with higher chromosomal phenotypic similarity is successful and variations in DNA content can result from changes in chromosome structure, leading to wide variations in morphology [55]. 

Considering the difference in the geographical origin of the studied genotypes and the rapid changes in the nuclear DNA amount in plants in response to environmental stimuli [56], differences in the amount of nuclear DNA were expected. According to previous results, the change in the relative amount of nuclear DNA in plants is probably due to the presence of abnormal chromosomes, the role of the amount of nuclear DNA in environmental adaptations [57], and changes in chromosome length [58]. Turkey is one of the most important centers of leguminous diversity with different climates, so it has good potential for the improved development of these plants, and it is necessary to conduct careful planning to use this diversity optimally. In order to increase the production and use of legumes, conventional plant breeding techniques have played the greatest role in their genetic improvement. Still, their breeding speed is lower than that of other crops, such as cereals, for some reason. The creation of synthetic cultivars requires germplasm evaluation, selection of superior parental genotypes, and knowledge of the genetic and cytogenetic parameters of traits [59,60].

## 5. Conclusions

It can be concluded that the results of the examination of these promising genotypes selected among the many genotypes used in different regions of Turkey show that the Van and Hakari genotypes have the same karyotypic formula, including eleven metacentric chromosomes. The shapes of their chromosomes are also similar, which can be considered for breeding programs between these two genotypes. In fact, cytogenetic studies are regarded as primary and fundamental achievements in breeding research, because determining the number and similarity of chromosomes and ploidy levels as well as the amount of the nuclear DNA content is essential to choosing the appropriate breeding method. On the other hand, the Bitlis and Elaziğ genotypes have close and similar karyotypes, which include ten metacentric chromosomes and one submetacentric chromosome. In addition, upon examining the amount of nuclear DNA content in these four genotypes, the highest amount was found in the Bitlis genotype, and the Hakari, Van, and Elaziq genotypes had lower values of this parameter. Knowledge of these parameters in this plant can be helpful for the breeders who decide to carry out the breeding of *Phaseolus* according to the conditions in these regions in Turkey, because successful breeding is possible when the characteristics of the size of the genome and the number of chromosomes in the researched plants are known.

## Figures and Tables

**Table 1 genes-14-00047-t001:** Genotype code numbers, names, locations, growing patterns, latitude, longitude, and altitude of the common bean used as material in the study.

Genotype Code Number	Genotype Name	Location	Growing Pattern	Latitude	Longitude	Altitude (m)
471-7	BT-52	Kuşlu köyü	Climbing	38°19′739″	42°14′841″	1615
471-12	BT-97	Kuştaşı köyü	Dwarf	38°29′645″	42°04′575″	2002
471-3	BT-123	Yazlikonak	Climbing	38°26′765″	42°51′660″	1810
471-9	BT-124	Yolalan köyü	Dwarf	38°17′889″	42°15′891″	1543
471-35	BT-114	Yumrumeşe köyü	Climbing	38°26′765″	42°51′660″	1659
471-20	BT-117	Cınarbaşi köyü	Dwarf	38°15′861″	42°17′972″	1710
471-17	BT-68	Mutki Kavakbası köyü	Dwarf	38°28′884″	41°48′924″	1303
471-28	BT-28	Kalkanlı köyü	Climbing	38°07′704″	42°37′670″	2004
471-39	BT-58	Topköy köyü	Dwarf	38°24′217″	42°16′295″	1752
471-10	BT-100	Sütderesi köyü	Dwarf	38°36′113″	42°00′993″	1307
596-11	HK-30	Bay köyü	Climbing	37°32′687″	42°43′333″	1832
596-27	HK-33	Otluca köyü 1	Climbing	37°36′105″	43°41′643″	2096
596-21	HK-8	Otluca köyü 2	Climbing	37°36′246″	43°42′370″	2054
596-32	HK-56	Otluca köyü 3	Dwarf	37°36′33″	43°42′525″	2095
596-3	HK-12	Ağaç dibi köyü	Climbing	37°29′370″	43°38′184″	2097
596-26	HK-18	Üzümlü köyü	Climbing	37°29′773″	43°34′389″	1135
596-18	HK-79	Çimenli köyü	Climbing	37°29′096″	43°37′693″	1137
596-36	HK-73	Şemdinli merkez	Climbing	37°19′045″	44°33′625″	1408
596-19	HK-71	Şemdinli güzel konak	Climbing	37°25′223″	44°29′056″	1724
596-25	HK-66	Yüksekova köprücü köyü	Climbing	37°34′196″	43°22′555″	1866
387-1	VN-26	Bahçesaray batalor köyü	Climbing	38°30′234″	42°23′285″	1958
387-33	VN-20	Bahçesaray aksaray köyü	Climbing	38°30′862″	42°19′043″	1684
387-34	VN-24	Bahçesaray çatbayır köyü	Climbing	38°30′943″	42°24′735″	1917
387-2	VN-58	Başkale barış köyü	Climbing	38°01′147″	43°39′146″	2224
387-38	VN-50	Gürpınar Merkez	Climbing	38°19′124″	44°33′625″	1748
387-13	VN-48	Çatak Merkez	Climbing	38°00′451″	43°03′619″	1502
387-40	VN-27	Çatak karsıyaka köyü	Climbing	38°00′721″	44°33′625″	1783
387-37	VN-30	Çaldıran Merkez	Climbing	37°42′409″	44°07′448″	2005
387-14	VN-28	Şemdinli Merkez	Climbing	37°19′045″	44°33′625″	2072
387-31	VN-39	Başkale germon köyü	Dwarf	37°59′908″	43°57′436″	2247
411-4	EL-11	Maden gezin Merkez	Dwarf	38°31′283″	89°31′880″	1266
411-24	EL-16	Sivrice kızıltepe köyü	Dwarf	38°28′865″	39°31′155″	1291
411-15	EL-40	Maden küçükova köyü	Dwarf	38°02′552″	39°32′526″	1410
411-23	EL-30	Maden Merkez	Dwarf	38°30′760″	39°33′172″	1350
411-5	EL-34	Maden yeşil ada köyü	Dwarf	38°32′905″	39°38′695″	1503
411-29	EL-33	Maden kardeldere köyü	Dwarf	38°33′108″	39°35′801″	1404
411-16	EL-28	Sivice elmasaray köyü	Dwarf	38°24′728″	39°23′341″	1364
411-30	EL-27	Sivice boşkaynak köyü	Dwarf	38°22′855″	39°22′217″	1390
411-6	EL-26	Sivrice kavak köyü	Dwarf	38°23′522″	39°25′121″	1304
411-22	EL-21	Sivrice kızıltepe köyü	Dwarf	38°29′472″	39°31′362″	1250

**Table 2 genes-14-00047-t002:** Variations in the karyotype features in examined *Phaseolus vulgaris* L. genotypes.

Genotype	MTCL ± SE (µm)	MCL ± SE (µm)	MCI ± SE (µm)	MAR ± SE	Karyotype Formula
B.T.	34.24 ± 0.35	3.11 ± 0.32	46.83 ± 3.52	1.15 ± 0.10	2n = 2x = 22 = 10 m + 1 sm
H.K.	33.98 ± 1.35	3.09 ± 0.12	45.71 ± 0.91	1.25 ± 0.05	2n = 2x = 22 = 11 m
VN	24.90 ± 1.12	2.26 ± 0.01	45.66 ± 1.32	1.26 ± 0.08	2n = 2x = 22 = 11 m
E.L.	15.65 ± 0.99	1.42 ± 0.10	43.18 ± 0.99	1.36 ± 0.07	2n = 2x = 22 = 10 m + 1 sm

MTCL, Mean Total Haploid Chromosome Length; MCL, Mean Chromosome Length; MCI, Mean Centromere Index; MAR, Mean Arm Ratio; SE, Standard Error; µm, micrometer; m, metacentric; sm, submetacentric. B.T., Bitlis; H.K., Hakkari; VN, Van; E.L., Elaziğ.

**Table 3 genes-14-00047-t003:** Nuclear DNA content and total diploid chromosome length in studied genotypes of *P. vulgaris* L.

Genotype	DNA Content (pg)	Total Diploid Chromosome Length (µm)
B.T.	3.28 ± 0.101	68.48 ± 0.012
H.K.	3.12 ± 0.142	67.96 ± 0.017
VN	3.05 ± 0.029	49.80 ± 0.014
E.L.	1.49 ± 0.032	31.30 ± 0.019

pg, picogram; µm, micrometer; B.T., Bitlis; H.K., Hakkari; VN, Van; E.L., Elaziğ.

**Table 4 genes-14-00047-t004:** Analysis of variance of TCL (Total Chromosome Length) in *P. vulgaris* L. genotypes.

S. O. V	df	MS
Genotypes	3	936.89 **
Error	8	0.68
Total	11	

** Significant at *p* < 0.01.

**Table 5 genes-14-00047-t005:** Mean comparison of different *P. vulgaris* L. genotypes studied.

Genotype	TCL (Total Chromosome Length) ± S.E. (µm)
B.T.	68.48 ± 0.012 a
H.K.	67.96 ± 0.017 a
VN	49.80 ± 0.14 b
E.L.	31.30 ± 0.019 c

The values with the same letters were not significant at *p* < 0.01 in Duncan’s Multiple Range Test. B.T., Bitlis; H.K., Hakkari; VN, Van; E.L., Elaziğ.

**Table 6 genes-14-00047-t006:** Analysis of variance of nuclear DNA content in *P. vulgaris* L. genotypes.

S. O. V	df	MS
Genotypes	3	2.095 **
Error	8	0.099
Total	11	

** Significant at *p* < 0.01.

**Table 7 genes-14-00047-t007:** Mean comparison of different *P. vulgaris* L. studied genotypes.

Genotype	Nuclear DNA Content Mean ± S.E. (pg)
B.T.	3.28 ± 0.101 a
H.K.	3.12 ± 0.142 a
VN	3.05 ± 0.029 a
E.L.	1.49 ± 0.032 b

The values with the same letters were not significant at *p* < 0.01 in Duncan’s Multiple Range Test. B.T., Bitlis; H.K., Hakkari; VN, Van; E.L., Elaziğ.

## Data Availability

The original contributions presented in the study are included in the article; further inquiries can be directed to the corresponding author/s.

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
