# Peer review of "Analysis of Nuclear DNA Content and Karyotype of Phaseolus vulgaris L."

_genes, 2022, doi:10.3390/genes14010047_

Round 1

Reviewer 1 Report

The paper sounds interesting and discusses the important aspects of DNA content and karyotype analysis of P. vulgaris with the help of Flow Cytometry, The paper is well written, but the authors should address few minor issues as suggested below -

1)                  Carefully check for typos, spacing issues, grammar usage and language,

2)                  You used Lewitsky solution in your study as fixative, but it was not mention about the concentration of this solution and the components of the solution. Please explain it.

3)                  In karyotype analysis of samples, what method has been used to determine the type of chromosomes?

4)                  At what time of day were the root tip samples taken?

5)                  It might be better to mention in the methods what time of day the sampling is done, so more metaphases can be observed.

6)                  Did the hydrolysis take place under wet conditions?

7)                  How many replications were used for analysis of metaphases?

8)                  Please replace centromere index with Centromeric Index.

9)                  Please replace arm ratio with Arm Ratio.

10)              In Lines “161 and 162” write “the mean chromosomes length” for “the chromosomes mean length”.

11)              In line “162” it is better to write “Total Length of chromosomes”

12)              In line “163” it is better to mention the format of chromosome pictures.

13)              In lines “176 and 177” please write explanation of MTCL and MCL.

14)              It is better to write one sentence for difference between MTCL and MCL.

15)              Please write specific formulas for CI and AR. There is no any information about them.

16)              In lines “147 and 148” from 8 provinces (Malatya, Elazig, Tunceli, Bingol, Mus, Bitlis, Van and Hakkari) … please mention which part of these provinces?

17)              In line “28” please replace were measured with “were calculated”.

18)              In lines “35-38” please write full name for regions.

19)              In line “153” please mention exact coordinates.

20)              In lines “71 and 72” please improve this sentence and write clearer.

21)              In line “74” it is better to written “the chromosome number and morphologyinstead of “the number and morphology of chromosomes”.

Author Response

Review 1 Report

The paper sounds interesting and discusses the important aspects of DNA content and karyotype analysis of P. vulgaris with the help of Flow Cytometry, the paper is well written, but the authors should address few minor issues as suggested below –

Authors response: With regards and many thanks for illustrative and constructive comments of respected reviewers. We tried to consider most of the reviewers’ comments as per suggested in the reports and listed as below:

  • Carefully check for typos, spacing issues, grammar usage and language,

Authors response: It was done in revised MS.

  • You used Lewitsky solution in your study as fixative, but it was not mention about the concentration of this solution and the components of the solution. Please explain it.

Authors response: This Solution was a mixture of 1% chromic acid and 10% formaldehyde.

  • In karyotype analysis of samples, what method has been used to determine the type of chromosomes?

Authors response: To determine the type and classification of chromosomes according to the location of the centromere, the Levan method (Levan et al., 1964) has been used.

  • At what time of day were the root tip samples taken?

Authors response: Sampling was done between 8 and 10 AM local time.

  • It might be better to mention in the methods what time of day the sampling is done, so more metaphases can be observed.

Authors response: As mentioned, the sampling of the tips of the roots of the plants was done between 8 and 10 AM local time, because during these hours the meristem of the root tip contains a large number of cells undergoing mitosis and the Mitotic Index is high.

  • Did the hydrolysis take place under wet conditions?

Authors response: Yes, the hydrolysis was done in a hot water bath.

  • How many replications were used for analysis of metaphases?

Authors response: 10 metaphases used as 10 replications.

  • Please replace centromere index with Centromeric Index.

Authors response: It was done in revised MS.

  • Please replace arm ratio with Arm Ratio.

Authors response: It was done in revised MS.

  • In Lines “161 and 162” write “the mean chromosomes length” for “the chromosomes mean length”.

Authors response: It was replaced.

  • In line “162” it is better to write “Total Length of chromosomes”

Authors response: It was corrected in revised MS.

  • In line “163” it is better to mention the format of chromosome pictures.

Authors response: The format of the chromosome pictures used is BMP.

  • In lines “176 and 177” please write explanation of MTCL and MCL.

Authors response: MTCL: Mean Total Chromosomes Length- MCL: Mean Chromosome Length.

  • It is better to write one sentence for difference between MTCL and MCL

Authors response: MTCL means Mean Total Chromosomes Length of whole nucleus chromosomes, and MCL means Mean one Chromosome Length.

  • Please write specific formulas for CI and AR. There is no any information about them.

Authors response: It was added in revised MS.

  • In lines “147 and 148” from 8 provinces (Malatya, Elazig, Tunceli, Bingol, Mus, Bitlis, Van and Hakkari) … please mention which part of these provinces?

Authors response: The central parts of the all provinces.

  • In line “28” please replace “were measured” with “were calculated”.

Authors response: It was replaced.

  • In lines “35-38” please write full name for regions.

Authors response: It was added.

  • In line “153” please mention exact

Authors response:

  • In lines “71 and 72” please improve this sentence and write clearer.

Authors response: It was changed in revised MS.

  • In line “74” it is better to written “the chromosome number and morphologyinstead of “the number and morphology of chromosomes”.

Authors response: It was done in revised MS.

Reviewer 2 Report

The manuscript genes-2018312 reports interesting findings on the karyotype variation and nuclear DNA content of four genotypes of Phaseolus vulgaris L. cultivated in Turkey. Karyotypic characteristics and the nuclear DNA content of the genotype were estimated using the flow cytometry method. For each genotype, karyotype parameters were measured, such as the number of somatic chromosomes, mean total chromosome length, mean centromere index, and mean arm ratio.

The study is of great interest and could add knowledge to the field. The experiment was conducted with valid methodologies. However, some parts in manuscript are not explained with sufficient details and need to be explained by the authors. See specific comments below.

Keywords: I suggest using terms different from those present in the title.

Introduction: The Introduction correctly places the study in the context with a clear statement of the purpose of the study. The description of the working hypotheses should be described more clearly.

Materials and Methods:  I would add missing description of some experiments:

·         In line “55” please specify which kind of adaptation to water.

·         Individual chromosomes can be affected differently from pressure applied on them during steps of the cytological preparations. Therefore, some chromosomes can be larger than their original size while some others can be smaller included in the same cell. All of these makes impossible to identify chromosomes ambiguously which limits the usage of the technique in karyotype and genome analyses. How can you manage this problem?

·         How chromosome measurements have been done?

·         Are the measurements taken from printed photos or from microscope photos?

·         Which device was used to estimate the amount of nuclear DNA content in researched genotypes?

·         Is the UV or laser method used in the nuclear DNA measurement device?

·         Line “132”: the sentence “It was reported …” does not fit into the paragraph or a liaison sentence is missing, please improve.

·         Line “147’: which method of selection was used?

·         In Method and Materials, passport data such as coordinates would be of great importance for readers and users.

·         Why did you used root tips in chromosome observation analysis?

·         In line 169, please write reference for the method.

·         Provide the composition of Lewitsky solution is better to.

·         How long were the samples pre-treated with 8-Hydroxiquinolin solution?

·         What was the pre-treatment concentration?

·         Is the size of the satellites considered when measuring the total length of chromosomes?

·         In staining, add Aceto-Orcein concentration.

·         Why did you choose Allium cepa for control?

·         It is better to specify the type of flow cytometry device because there are different types of this device.

·         Line “85”, “chromosomal status…” rephrase the sentence as it does not make a lot of sense and the sentence may be placed in the discussion paragraph.

·         Please write a reference to chromosome measurement program.

Results: The results description is clear and supported by appropriate figures and tables

DiscussionAuthors correctly discussed the results from the perspective of previous studies, considering a valid number of references. The concept flow is clear.

Other comments: I would add a Conclusion section. The latter should contain the importance of the findings obtained.

Author Response

Reviewer 2

The manuscript genes-2018312 reports interesting findings on the karyotype variation and nuclear DNA content of four genotypes of Phaseolus vulgaris L. cultivated in Turkey. Karyotypic characteristics and the nuclear DNA content of the genotype were estimated using the flow cytometry method. For each genotype, karyotype parameters were measured, such as the number of somatic chromosomes, mean total chromosome length, mean centromere index, and mean arm ratio.

The study is of great interest and could add knowledge to the field. The experiment was conducted with valid methodologies. However, some parts in manuscript are not explained with sufficient details and need to be explained by the authors. See specific comments below.

  1. Keywords:I suggest using terms different from those present in the title.

Authors response: It was corrected in the revised MS.

  1. Introduction:The Introduction correctly places the study in the context with a clear statement of the purpose of the study. The description of the working hypotheses should be described more clearly.

Authors response: It was described in revised MS.

Materials and Methods:  I would add missing description of some experiments:

  1. In line “55” please specify which kind of adaptation to water.

Authors response:

  1. Individual chromosomes can be affected differently from pressure applied on them during steps of the cytological preparations. Therefore, some chromosomes can be larger than their original size while some others can be smaller included in the same cell. All of these makes impossible to identify chromosomes ambiguously which limits the usage of the technique in karyotype and genome analyses. How can you manage this problem?

Authors response: Yes, exactly. For this reason, we increased the number of replications to reduce the standard error, and 10 replications were considered.

  1. How chromosome measurements have been done?

Authors response: Chromosome measurements have been done with Micro measure 3.3.

  1. Are the measurements taken from printed photos or from microscope photos?

Authors response: Microscope photos.

  1. Which device was used to estimate the amount of nuclear DNA content in researched genotypes?

Authors response: PARTEC Flow cytometry

  1. Is the UV or laser method used in the nuclear DNA measurement device?

Authors response: Laser

  1. Line “132”: the sentence “It was reported …” does not fit into the paragraph or a liaison sentence is missing, please improve.

Authors response: It was changed in revised MS.

  1. Line “147’: which method of selection was used?

Authors response: Mass selection

  1. In Method and Materials, passport data such as coordinates would be of great importance for readers and users.

Authors response:

  1. Why did you used root tips in chromosome observation analysis?

Authors response: Because in cytogenetic studies, to find more metaphases, the easiest and best example is to use the root tip meristem.

  1. In line 169, please write reference for the method.

Authors response: It was added.

  1. Provide the composition of Lewitsky solution is better to.

Authors response: This Solution was a mixture of 1% chromic acid and 10% formaldehyde.

  1. How long were the samples pre-treated with 8-Hydroxiquinolin solution?

Authors response: 36 hours

  1. What was the pre-treatment concentration?

Authors response: It was done inside the refrigerator (4°C).

  1. Is the size of the satellites considered when measuring the total length of chromosomes?

Authors response: Yes, it considered in total length of chromosomes.

  1. In staining, add Aceto-Orcein concentration.

Authors response: %2

  1. Why did you choose Allium cepa for control?

Authors response: Because of the similarity of the genome of this plant with Phaseolus genome

  1. It is better to specify the type of flow cytometry device because there are different types of this device.

Authors response: PARTEC Laser base Flow cytometry

  1. Line “85”, “chromosomal status…” rephrase the sentence as it does not make a lot of sense and the sentence may be placed in the discussion paragraph.

Authors response:  It was changed in the revised MS.

  1. Please write a reference to chromosome measurement program.

Authors response: It was added in revised MS.

  1. Results:The results description is clear and supported by appropriate figures and tables

Authors response:

  1. Discussion: Authors correctly discussed the results from the perspective of previous studies, considering a valid number of references. The concept flow is clear.

Authors response:

  1. Other comments: I would add a Conclusion The latter should contain the importance of the findings obtained.

Authors response: Conclusion section was added in revised MS.

Reviewer 3 Report

The manuscript entitled “Analysis of Nuclear DNA content and karyotype of Phaseolus vulgaris L.” by Kualaz et al. reports nuclear DNA content and karyotype variation in four P. vulgaris genotypes from Turkey. The authors used the flow cytometry method and measured various parameters. This study has no specific objective except for applying this technique to study the nuclear DNA and karyotypic characteristics of four varieties. I have the following comments.

1.       The abstract does not provide any hypothesis or statement that indicates the need for the study.

2.       The abstract lacks and conclusion statement. The authors did not add any sentence that can tell us what is the utility of this work.

3.       To my knowledge, keywords should be different from the title.

4.       L37-63. Did not establish any rationale for the study or the need for the study. It does not provide any background information on the topic except that legumes are important and that we are facing drought stress in legumes. The authors failed to relate this information to the objective of the study.

5.       Authors should add only selective and most appropriate literature for the INTRODUCTION section. How these parameters were previously used to determine variation between varieties or cultivars? mostly they are talking about interspecific and intraspecific. But we need here the literature that explains how these characteristics are linked with diversity and what are successful stories in this regard.

6.       The most relevant literature in the introduction section starts from L97. Previous paragraphs are just not useful and should be highly reduced to avoid irrelevant material.

7.       L134. Do not repeat the full sci name of the species. Also, it needs to be italicized. This mistake is repeated multiple times.

8.       L146… Authors must provide any reference that says these four lines/genotypes are the best. What are the criteria for saying them the best?

9.       I do not understand the objective of this study. Why do the authors want to study the nuclear DNA and karyotyping of only four accessions/genotypes? What will this study solve in terms of biological or commercial or breeding benefits? The RESULTS section is just a description of the numbers and data but failed to tell what it actually means and how it is useful for the researchers.

10.   The DISCUSSION section is too weak, it distracts the reader. It gives no reason for the utility of the given results. Only the authors drag the words and tell that it is confirming previous findings of this and that study. So, if it confirms any study, what is actually adding to the field/subject? How breeders can benefit from this information? How it helps us understand the differences in the four varieties used? What did your study conclude? 

Reviewer 3 Report

The manuscript entitled “Analysis of Nuclear DNA content and karyotype of Phaseolus vulgaris L.” by Kualaz et al. reports nuclear DNA content and karyotype variation in four P. vulgaris genotypes from Turkey. The authors used the flow cytometry method and measured various parameters. This study has no specific objective except for applying this technique to study the nuclear DNA and karyotypic characteristics of four varieties. I have the following comments.

Thanks for your very deep consideration and comments.

  1. The abstract does not provide any hypothesis or statement that indicates the need for the study.

Authors response: The abstract was revised per your comments.

  1. The abstract lacks and conclusion statement. The authors did not add any sentence that can tell us what the utility of this work is.

Authors response: It was added to the abstract in the revised manuscript.

  1. To my knowledge, keywords should be different from the title.

Authors response: It was corrected in the revised MS.

  1. L37-63. Did not establish any rationale for the study or the need for the study. It does not provide any background information on the topic except that legumes are important and that we are facing drought stress in legumes. The authors failed to relate this information to the objective of the study.

Authors response: In the revised version, we tried to correct most of the parts according to your useful comments so that we can relate the obtained information to the objective of the study. I hope we have been able to apply your opinion.

  1. Authors should add only selective and most appropriate literature for the INTRODUCTION section. How these parameters were previously used to determine variation between varieties or cultivars? mostly they are talking about interspecific and intraspecific. But we need here the literature that explains how these characteristics are linked with diversity and what are successful stories in this regard.

Authors response: We tried to change and improve parts of the introduction.

  1. The most relevant literature in the introduction section starts from L97. Previous paragraphs are just not useful and should be highly reduced to avoid irrelevant material.

Authors response: Many of the unnecessary parts that you mentioned have been removed.

  1. Do not repeat the full sci name of the species. Also, it needs to be italicized. This mistake is repeated multiple times.

Authors response: It was revised.

  1. L146… Authors must provide any reference that says these four lines/genotypes are the best. What are the criteria for saying them the best?

Authors response: The desired information was added to the relevant section.

  1. I do not understand the objective of this study. Why do the authors want to study the nuclear DNA and karyotyping of only four accessions/genotypes? What will this study solve in terms of biological or commercial or breeding benefits? The RESULTS section is just a description of the numbers and data but failed to tell what it actually means and how it is useful for the researchers.

Authors response: The objectives and advantages of the study that may be useful for researchers were added as “Conclusion”, which I hope has improved the MS.

  1. The DISCUSSION section is too weak, it distracts the reader. It gives no reason for the utility of the given results. Only the authors drag the words and tell that it is confirming previous findings of this and that study. So, if it confirms any study, what is actually adding to the field/subject? How breeders can benefit from this information? How it helps us understand the differences in the four varieties used? What did your study conclude? 

Authors response: Discussion section has been improved in the light of present literature. Significance of outcome have been added to strengthened this part.

Round 2

Reviewer 3 Report

1.       Authors have modified the title and abstract (from the second statementL27), which is too general and indicates that probably they are doing it for the first time in this species. However, this is not novel and reports on the karyotype and nuclear DNA content have been published in the last two decades. Thus, changing the title and abstract clearly indicates that the authors have no specific objective here. Previously they claimed the study was for diversity, and now they added that “Appropriate and stable cytological methods for each plant facilitate and make……”.

2.       Again in L87 and onward, they write that it is necessary to have complete and sufficient information …….. The authors did not write that the scientific community have already this information for Phaseolus vulgaris. They failed to highlight why only the selected accessions need these data and how it is urgently needed for breeding.

3.       L103. No, here we are not talking about classification problems. The species has already been classified and there exist no such issue and need.

4.       L159. Authors hypothesize that the selected accessions “may be different in terms of …………”. On what basis they hypothesize this? No clue here.

5.       In 2.1. section. No clear data, references, or reports about these four accessions were presented. Neither their phenotypic description nor their performance and no figure. How would the readers know that these four accessions have these traits? A brief comparative data on a few traits and morphology could have made this selection confident. But such information is missing.

6.       Does this manuscript stand as a research paper at all? It could have been a short communication or some other category.

7.       No changes in discussion as per earlier comments.

Author Response

Dear Editor

Please accept my apologies for the late reply. In the second round, all parts of the manuscript were tried to be rewritten and revised based on constructive and useful comments of the third reviewer. Also, the recent new references were added to the necessary parts. I hope that we have been able to fix the previous mistakes and lacks in the new revised version. We tried to consider most of the reviewers’ comments as suggested in the reports below:

Reviewer 3 Report Round 2

  • The authors have modified the title and abstract (from the second statement, L27), which is too general and indicates that they are probably doing it for the first time in this species. However, this is not novel, and reports on the karyotype and nuclear DNA content have been published in the last two decades. Thus, changing the title and abstract indicates that the authors have no specific objective. Previously, they claimed the study was for diversity, and now they added, "Appropriate and stable cytological methods for each plant facilitate and make……”.

Authors response: First of all, thanks for your full attention to the different parts of our manuscript. It should be mentioned that our team has started a project of collecting, evaluating, separating, and starting breeding programs in common beans for several years. This study is a small part of this big project. Molecular analysis of chromosomes (FISH), as well as evaluation of the correlation between genotypes and climate parameters, has also started. I hope that all changes made in different parts of the manuscript will be satisfied and improve the manuscript's quality.

  • Again in L87 and onward, they write that it is necessary to have complete and sufficient information …….. The authors did not write that the scientific community already has this information for Phaseolus vulgaris. They failed to highlight why only the selected accessions need these data and how it is urgently needed for breeding.

Authors response: In the revised manuscript, it was tried to write most of the previous works that were carried out on the cytogenetic analysis of common beans as well as clarify the need for the data evaluated in this study.

  • No, here we are not talking about classification problems. The species has already been classified, and there exists no such issue and need.

Authors response: The comments were considered and corrected in the revised version.

  • Authors hypothesize that the selected accessions “may be different in terms of …………”. On what basis do they hypothesize this? No clue here.

Authors response: The hypothesis was considered and corrected in the revised version.

  • In 2.1. section. No clear data, references, or reports about these four accessions were presented. Neither their phenotypic description nor their performance and no figure. How would the readers know that these four accessions have these traits? A brief comparative data on a few traits and morphology could have made this selection confident. But such information is missing.

Authors response: Section 2.1 was rewritten, and a table including plant samples, code number, accession name, location, and growing pattern, as well as their geographical information, including latitude, longitude, and altitude, was added. This study was designed based on previous studies carried out by one of the authors, so the studied genotypes of Phaseolus were selected based on previous phenotypic properties and performance. So, the revised version has tried clarifying the previous ambiguities by referring to this issue.

  • Does this manuscript stand as a research paper at all? It could have been a short communication or some other category.

Authors response: By revision and modification of the manuscript, especially ANOVA and statistical grouping, it can be proposed as a research paper.

  • No changes in the discussion as per earlier comments.

Authors response: This section was rewritten, and it was tried to compare the results of the present study with the results of the karyological studies conducted before on common beans.
